# Genome-Wide Analysis of Prognostic Alternative Splicing Signature and Splicing Factors in Lung Adenocarcinoma

**DOI:** 10.3390/genes11111300

**Published:** 2020-10-31

**Authors:** Ya-Sian Chang, Siang-Jyun Tu, Hui-Shan Chiang, Ju-Chen Yen, Ya-Ting Lee, Hsin-Yuan Fang, Jan-Gowth Chang

**Affiliations:** 1Epigenome Research Center, China Medical University Hospital, 404 Taichung, Taiwan; t25074@mail.cmuh.org.tw (Y.-S.C.); t24399@mail.cmuh.org.tw (J.-C.Y.); t23701@mail.cmuh.org.tw (Y.-T.L.); 2Department of Laboratory Medicine, China Medical University Hospital, 404 Taichung, Taiwan; t34752@mail.cmuh.org.tw (S.-J.T.); D18448@mail.cmuh.org.tw (H.-S.C.); 3Center for Precision Medicine, China Medical University Hospital, 404 Taichung, Taiwan; 4Department of Medical Laboratory Science and Biotechnology, China Medical University, 404 Taichung, Taiwan; 5Department of Thoracic Surgery, China Medical University Hospital, 404 Taichung, Taiwan; d17573@mail.cmuh.org.tw; 6School of Medicine, China Medical University, 404 Taichung, Taiwan; 7Department of Bioinformatics and Medical Engineering, Asia University, 413 Taichung, Taiwan

**Keywords:** lung adenocarcinoma, RNA sequencing, alternative splicing, splicing factor

## Abstract

Analysis of The Cancer Genome Atlas data revealed that alternative splicing (AS) events could serve as prognostic biomarkers in various cancer types. This study examined lung adenocarcinoma (LUAD) tissues for AS and assessed AS events as potential indicators of prognosis in our cohort. RNA sequencing and bioinformatics analysis were performed. We used SUPPA2 to analyze the AS profiles. Using univariate Cox regression analysis, overall survival (OS)-related AS events were identified. Genes relating to the OS-related AS events were imported into Cytoscape, and the CytoHubba application was run. OS-related splicing factors (SFs) were explored using the log-rank test. The relationship between the percent spliced-in value of the OS-related AS events and SF expression was identified by Spearman correlation analysis. We found 1957 OS-related AS events in 1151 genes, and most were protective factors. Alternative first exon splicing was the most frequent type of splicing event. The hub genes in the gene network of the OS-related AS events were *FBXW11*, *FBXL5*, *KCTD7*, *UBB* and *CDC27*. The area under the curve of the MIX prediction model was 0.847 for 5-year survival based on seven OS-related AS events. Overexpression of SFs *CELF2* and *SRSF5* was associated with better OS. We constructed a correlation network between SFs and OS-related AS events. In conclusion, we identified prognostic predictors using AS events that stratified LUAD patients into high- and low-risk groups. The discovery of the splicing networks in this study provides an insight into the underlying mechanisms.

## 1. Introduction

Lung cancer is one of the most common malignant neoplasms worldwide with 2.1 million new cases and 1.8 million deaths in 2018 [1]. There are many risk factors for the development of lung cancer, including tobacco smoking, diet, alcohol consumption, ionizing radiation, occupational exposure, air pollution and chronic inflammation from infections and other medical conditions. Genetic susceptibility is also a lung cancer risk factor [2]. Non-small-cell lung cancer (NSCLC) accounts for approximately 80–85% of lung cancer cases; lung adenocarcinoma (LUAD) and lung squamous cell carcinoma are the two major pathological subtypes of NSCLC [3,4]. Numerous lung cancer genomic studies have shown hot gene mutations, such as in the *EGFR*, *TP53*, *KRAS* and *BRAF* genes, and these driver genes have shown ethnic differences [5,6]. Although much effort has been spent developing new molecular targeted therapies and immunotherapy agents, the 5-year survival rate of patients with lung cancer is still low. Therefore, it is vital to identify new prognostic markers to develop personalized treatments for patients with lung cancer.

Alternative splicing (AS) plays an important role in the post-transcriptional process that generates multiple transcripts from the same genes, resulting in protein diversity. More than 95% of human genes undergo AS in physiological processes [7], and aberrant RNA splicing may play an important role in driving cancer development and progression [8,9] by influencing the metabolism, apoptosis, cell cycle control, invasion, metastasis and angiogenesis of cancer cells and their microenvironment [10]. Furthermore, aberrant splice variants are believed to have an effect on the efficacy of targeted therapies, chemotherapy, hormone therapy or immunotherapy. Some examples of these are the *TP53* splice variant (∆40p53), which was linked to cisplatin resistance [11], and the *BRAF* V600E (p61 *BRAF* V600E) splice variant, which was linked to vemurafenib resistance [12]. Many aberrant splicing events have been discovered in lung cancer, such as of *BCL2L1*, *MDM2*, *MDM4*, *MET* and *NUMB* [13]. 

Splicing factors (SFs) are regulators of AS events, and changes in the expression and/or mutation of SFs can alter the splicing of oncogenes and tumor suppressors [14]. Thus, aberrant AS and SFs are considered to be defining hallmarks of cancer. In lung cancer, misregulation of these proteins has been reported [13].

Most studies have used The Cancer Genome Atlas (TCGA) dataset to analyze AS events in different tumor types [15,16,17,18]. In RNA sequencing (RNA-seq) data from TCGA, splicing events were shown to serve as prognostic biomarkers in NSCLC [18]. 

Few studies have used both TCGA and the authors’ case-studies in a genome-wide comprehensive analysis of AS events and SFs of LUAD.

In this study, we conducted a systematic analysis of AS events and SFs in LUAD using our RNA-seq data and identified overall survival (OS)-related markers. Our results provide new insights for predicting the prognosis and evaluating clinical outcomes in LUAD patients and describe the underlying mechanisms.

## 2. Materials and Methods

### 2.1. Patient Samples

Tissue specimens were obtained from 40 Taiwanese patients with LUAD who underwent surgical resection from May 2007 to April 2014 at the China Medical University Hospital. Surgically resected specimens were grossly dissected and preserved immediately in liquid nitrogen following surgery. All subjects signed the written informed consent forms. The study was conducted in accordance with the Declaration of Helsinki, and the protocol was approved by the Institutional Review Board of the China Medical University Hospital (CMUH106-REC1-053). 

### 2.2. RNA-Seq

NucleoSpin RNA Kit (Macherey-Nagel GmbH, Düren, Germany) was used to extract total RNA from the clinical tissue samples according to the manufacturer’s instructions. We analyzed the quality, quantity and integrity of total RNA using a NanoDrop 1000 spectrophotometer and an Agilent 2100 Bioanalyzer (Agilent Technologies, Santa Clara, CA, USA). Samples with an RNA integrity >6.0 were used for RNA-seq. An mRNA-focused, barcoded library was generated using the TruSeq Stranded mRNA Library Preparation Kit (Illumina, San Diego, CA, USA). The libraries were sequenced on an Illumina Nova Seq 6000 (Illumina), using 2 × 151-bp paired-end sequencing flow cells according to the manufacturer’s instructions.

### 2.3. RNA-Seq Data Analysis

Trimmomatic was used for quality control of the RNA-seq data (Illumina) [19]. The reads underwent quality check steps such as the quality mean (AVG (q20)), sliding window trimming (SLIDINGWINDOW:4:15), head and tail trimming (LEADING:3, TRAILING:3), read length selection (MINLEN:100) and adaptive quality trimmer (MAXINFO:10:0.2). The passed reads were aligned to the human genome (GRCh38, Ensembl 84) via HISAT2 and assembled to transcripts by StringTie [20,21]. To quantify RNA expression of the SFs, featureCounts was used with Gencode v22 annotation without the mitochondrial genes.

### 2.4. Bioinformatics Identification of AS Events

We modified the high-confidence transcript determining method, based on Li et al. [22]. All alignment results were reannotated by merged GTF files, which were produced from the StringTie merge option with the original annotated samples. High-confidence transcripts were determined under two conditions: (1) with more than one exon and (2) where the fragments per kilobase of exon model per million reads mapped (FPKM) expression value was higher than 0.1 in at least one sample.

We used SUPPA2 to analyze the AS profiles and evaluate the mRNA splicing patterns from the high-confidence transcripts [23]. The percent spliced-in (PSI) value, which ranged from 0 to 1, was used to quantify the AS events and calculate the seven types of AS events as follows: retained intron (RI), skipping exon (SE), alternative 5’/3’ splice site events (A5/A3), alternative first/last exons (AF/AL) and mutually exclusive exons (MX). The ‘NA’ in the PSI matrix was replaced by 0 because expression of the isoform was 0. 

### 2.5. Construction of Prognostic Models and SF Correlation Network

A univariate Cox regression was used to analyze the relationship between AS events and the OS. AS events with a p-value <0.05, a standard error of the coefficient ≤0.8 and a 95% confidence interval ≤10 were regarded as OS-related events. 

Next, we grouped the OS-related events by AS type into AL, AF, A3, A5, MX, RI and SE and performed a multivariate Cox regression for each of the AS-type groups. The forward stepwise regression model was chosen. A mixture (MIX) multivariate Cox regression was constructed from the selected events from the multivariate Cox results of the seven AS-type groups. Prognostic models of the MIX and the seven AS type groups were constructed based on the following risk score formula: $$Riskscore_{type}=\sum^{\;}\left(\beta_{i}*psi_{i}\right)$$
where type includes AL, AF, A5, A3, MX, RI, SE and MIX; i indicates the AS event in each type group; βi is the coefficient value of event i from the multivariate Cox result; and $psi_{i}$ is the PSI value of event i.

We used the median score to divide the patients into high- and low-risk groups. A log-rank test was performed, and a Kaplan–Meier plot was created for each model. A time-dependent receiver operating characteristic (ROC) analysis was applied to compare the efficiency of each prediction model by the survival ROC package (version 1.0.3) [24].

The SF genes were obtained from SpliceAid2 [25]. The median value of the expression transcripts per million (TPM) value was used to divide the sample into high- or low-risk groups to perform the log-rank test. Spearman’s correlations between the expression of OS-related SFs and the PSI values from the top 20 splicing events in each type were visualized using Cytoscape 3.7 and Matplotlib [26,27].

### 2.6. Integrative Bioinformatics and Statistical Analysis

UpSetPlot 0.4.0 in Python was used to show the intersections between the genes and the seven AS events [28]. We then applied Lifelines in the univariate Cox regression, performed a log-rank test and created a Kaplan–Meier plot [29]. No correction was applied for multiple testing in the Cox regression analysis. SPSS software was used for the forward stepwise multivariate Cox regression analysis.

## 3. Results

### 3.1. OS-Related AS Events in LUAD

A flowchart of our study design is shown in Figure 1. Initially, 96,124 AS events were identified using SUPPA2. We applied univariate Cox regression analysis of OS to evaluate the prognostic impact of each AS event and to explore the prognostic value of the AS events in LUAD patients. Consequently, 1957 AS events in 1151 genes were shown to be significantly associated with the OS of LUAD patients (*p* < 0.05) (Appendix A). The AS events included 1170 AFs in 624 genes, 285 ALs in 214 genes, 230 SEs in 221 genes, 103 A5s in 97 genes, 89 A3s in 87 genes, 52 MXs in 50 genes and 28 RIs in 25 genes (Figure 2A). Among them, 80 AF, 25 SE, 10 AL, 2 A5 and 2 A3 genes were the same as in TCGA database (Figure 2B). The top 20 significant OS-related AS events within the seven types of AS are shown in Figure 3A–G. Most of these AS events were favorable AS prognostic factors, and only three AS events were adverse factors (two *ABHD14A* AS events and one *TMEM132E* AS event). We also found that one gene could have multiple OS-related AS events in LUAD. Thus, a subset of the overlapping AS events among the seven types of AS in LUAD is shown using an UpSetPlot diagram (Figure 4A). Notably, some of the OS-related AS genes underwent at least two types of AS, and a few even had more than two types of AS events. For example, *TANGO2* had up to four OS-related events (AF, SE, AL and MX). We used Cytoscape to generate gene interaction networks to explore the functional relationships among these significant OS-related events. We found that the survival-associated genes were related to the hub genes in the network, such as *FBXW11*, *FBXL5*, *KCTD7*, *UBB* and *CDC27* (Figure 4B).

### 3.2. Prognostic Predictors for the LUAD Cohort

We selected OS-related AS events as potential candidates to detect independent prognostic factors in LUAD patients. We separately performed multivariate Cox regression models for the independent prognostic factors for the seven types of candidate AS events. In our data analysis of each type of splicing pattern, all seven prognostic models constructed using the different types of AS events showed significant power to predict the outcomes of LUAD patients (Figure 5A–G). Furthermore, the candidate AS events from the seven predictors were combined to construct the MIX prognostic predictors. Notably, the MIX prognostic predictors also showed significant power to predict the outcomes (Figure 5H). The genes involved in all models are listed in Appendix A. The ROC curves were plotted, and the AUCs were calculated for each model to identify the most effective predictive model (Figure 5I). The AUC of the ROC for the A5 prognostic predictor was 0.907, followed by the SE and AL models with AUC scores of 0.905 and 0.893, respectively. The A5 events were annotated through Cytoscape analysis. We found that *NRP2* and *ADA2*
**(*CECR1*)** were involved in the heparin-binding pathway (Appendix A).

### 3.3. The Regulation Network of SFs and the Top 20 Significant OS-Related AS Events for the Seven Types of AS

A total of 71 SFs were obtained from the SpliceAid2 database (www.introni.it/spliceaid.html). We conducted survival analysis of the SFs based on gene expression to determine which SFs were associated with OS-related AS events in LUAD. The top five significant OS-related SFs were *CELF2*, *SRSF5*, *HNRNPK*, *ELAVL4* and *HNRNPC* (*p* < 0.1), and Kaplan–Meier curves showed that high expression of these SFs correlated with good outcomes (Figure 6A). 

The correlation between the PSI values of the most significant AS events and the expression of OS-related SFs was determined using Spearman’s test, and Cytoscape was used to visualize the network (Figure 6B and Appendix A). We identified 13 negative interactions in three adverse survival prognostic AS events and five SFs, whereas there were more positive interactions in favorable prognostic AS events and SFs. The top three correlations between SFs and OS-related AS events are shown in Figure 6C. 

## 4. Discussion

We identified AS events and regulatory SFs through an analysis of our LUAD cohort to explore the clinical significance of differential RNA splicing patterns. We detected 1957 OS-related AS events in 1151 genes using univariate Cox regression analysis. All of the top 20 significant OS-related AS events in AF, AL, SE, A5, A3 and MX were favorable prognostic factors (hazard ratio (HR) < 1). Only 3 of the top 20 significant OS-related AS events in RI were adverse prognostic factors (two *ABHD14A* AS events and one *TMEM132E* AS event) (HR > 1). In our study, the *ABHD14A*_ENST00000637025 and *TMEM132E*_ENST00000631683 isoforms were associated with poor prognosis. There was only one report of *ABHD14A* in cancer where it was downregulated in breast cancer with visceral organ metastasis [30]; however, the *ABHD14A* and *TMEM132E* AS events were not fully explored. From our AS study of *ABHD14A*, we showed that ENST00000637025 disrupted the function of ABHD14A, which is similar to the downregulation of *ABHD14A* and resulted in poor prognosis. The sequence reads of splicing isoforms of *ABHD14A* (ENST00000637025 and ENST00000635937) are shown in Appendix A. The AS of *TMEM132E* had a similar effect on TMEM132E, and we suggest that this protein may function as a tumor suppressor in the development of lung cancer; however, further studies are needed to confirm our suggestion.

One event of the tumor suppressor gene *CDKN2A* in the top 20 significant OS-related AS events in A5 was a favorable prognostic factor. The *CDKN2A*_ENST00000497750 isoform was associated with good prognosis. *CDKN2A* is known to be a negative regulator of the cell cycle, and its transcriptional variants (*p16INK4a*, *p14ARF* and *p12*) showed different inhibitory effects on the A549 human lung cancer cell line [31]. However, the *CDKN2A*_ENST00000497750 isoform has not been fully investigated to date, and we analyzed the functional change in A5 of *CDKN2A* by bioinformatics analysis and found this form will increase the function of the ankyrin repeat-containing domain. Further studies are required to confirm our preliminary results.

The gene interaction network was established based on the 1957 OS-related AS events using Cytoscape, and *FBXW11*, *FBXL5*, *KCTD7*, *UBB* and *CDC27* were identified as hub genes in the gene network. FBXW11 (also known as β-transducin repeat-containing protein 2) is an F-box protein of the ubiquitin–proteasome system (UPS). It can target various substrates for degradation to regulate cell proliferation and survival in cancer. The role of FBXW11 in tumorigenesis is not clear. Reduced expression of FBXW11 was reported in NSCLC tissues, and this protein was recognized as a tumor suppressor [32]. FBXL5 is also an F-box protein of the UPS. Previous studies demonstrated that FBXL5 functions as a tumor suppressor in gastric and cervical cancers [33,34]. However, Yao et al. reported that FBXL5 functions as an oncogene in colon cancer [35]. From our results, we suggest that these hub genes might be potential targets for the prevention and treatment of LUAD in the future. 

Interestingly, most of the top 20 OS-related AS events of the seven splicing patterns indicate a good prognosis (137 events showed favorable prognostic factors and only 3 events had adverse prognostic factors). Our finding agrees with a previous study in which it was reported that most OS-related AS events in LUAD were favorable prognostic factors (85 and 55 events were favorable and adverse prognostic factors, respectively) [18]. 

Further analysis of the prediction model created by one type of AS pattern showed that A5 events were more effective for distinguishing the survival outcome of LUAD patients than the predictor models built using the other six types of AS pattern. Notably, seven prognostic prediction models performed well, with an AUC > 0.81. In addition, the MIX integrated predictive model had an AUC of 0.847, and the Kaplan–Meier plot clearly separated the patients into high- and low-risk groups. Zhou et al. reported that the AUC of the ROC curve for the final prediction model constructed using eight AS events was 0.844 in males and 0.907 in females [36]. In our study, the MIX prediction model was constructed using eight OS-related AS events, and the number of events was identical.

We also identified five OS-related SFs, and the Kaplan–Meier plots of two of these SFs significantly distinguished between the high- and low-risk patients. All the top five OS-related SFs were protective factors, and high expression levels were associated with better prognosis. A splicing correlation network revealed that most of the protective AS events were positively correlated with the expression of SFs, whereas the risk AS events were negatively correlated. The serine–arginine-rich splicing factors (SRSFs) and heterogeneous nuclear ribonucleoprotein (HNRNP) proteins belong to two RNA SF families that have been extensively investigated [37]. SRSF and HNRNP proteins always show opposite functions during the process of alternative mRNA splicing. The SRSF protein binds sequence motifs that are associated with splicing enhancers, while the HNRNP protein tends to bind sequence motifs associated with splicing silencers. In this study, the expression of *CELF2*, *SRSF5*, *HNRNPK*, *ELAVL4* and *HNRNPC* influenced the prognosis of our LUAD patients, although the p-values of three SFs were higher than the standard significant p-value. We suggest that this was the result of simultaneous interactions between the splicing enhancers and silencers on the same gene and at the same or different sites. We analyzed TCGA data and whole-exome sequencing data of our cases, and the results showed that the top OS-associated SFs are not frequently mutated in LUAD. In our cases, there was *SRSF5* or *HNRNPC* mutation in only one person. There were no mutations in other genes in our cases. In TCGA data, the mutation frequencies of *CELF2*, *SRSF5*, *HNRNPK*, *ELAVL4* and *HNRNPC* were 1.76 (10/567), 0.53 (3/567), 1.06 (6/567), 3.17 (18/567), 1.06 (6/567), respectively. Moreover, *SRSF1* expression was not correlated with better survival in LUAD (Appendix A), which was different from the study of Coomer et al. [13].

## 5. Conclusions

In conclusion, our findings improve our understanding of the association between AS events and LUAD, this study had some limitations. First, the study was conducted based on our LUAD cohort, which was not a big sample size. Second, we did not perform the experiment to confirm variant function in LUAD patients. We need to clarify these limitations in the near future.

## Figures and Tables

**Figure 1 genes-11-01300-f001:**
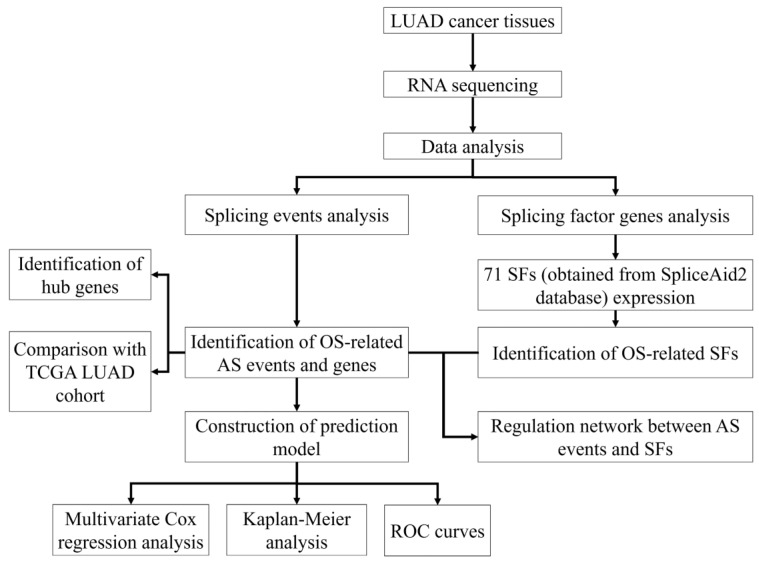
Flowchart of the study design.

**Figure 2 genes-11-01300-f002:**
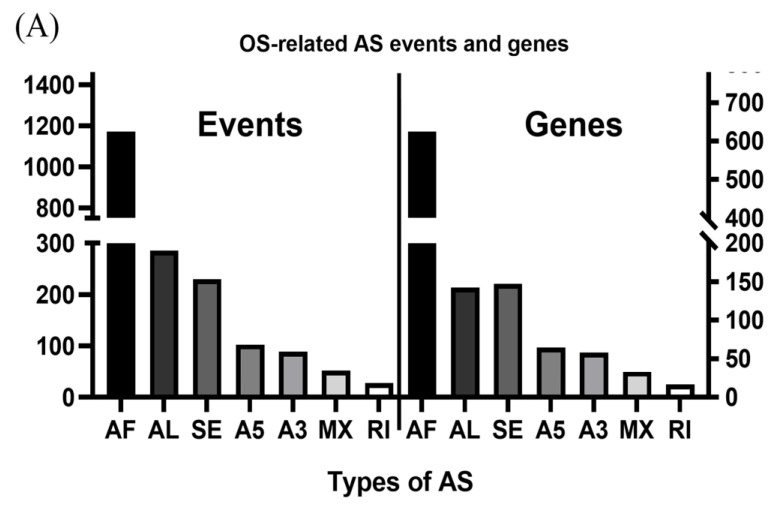
Overview of overall survival (OS)-related alternative splicing (AS) events. (**A**) The number of the seven types of OS-related AS events and genes involved in our lung adenocarcinoma (LUAD) cohort. (**B**) Venn diagrams showing the interrelationships between genes for LUAD in Taiwanese patients and The Cancer Genome Atlas (TCGA) databank. AF/AL: alternative first/last exons; SE: skipping exon; A5/A3: alternative5’/3’ splice site events; MX: mutually exclusive exons; RI: retained intron

**Figure 3 genes-11-01300-f003:**
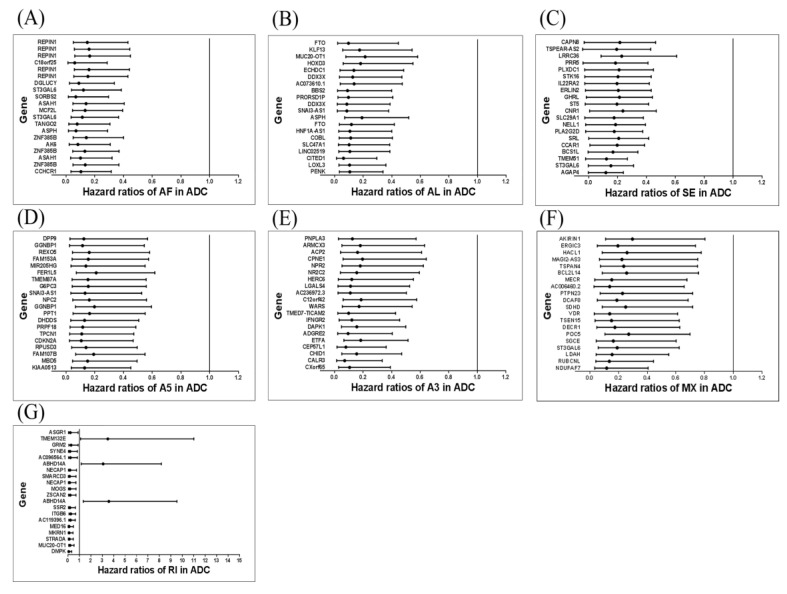
Forest plots for OS-related AS events in each subgroup. (**A**–**G**) Forest plots of hazard ratios (HRs) for the top 20 OS-related AS events in retained intron (RI), skipping exon (SE), alternative 5’ splice site events (A5), alternative 3’ splice site events (A3), alternative first exons (AF), alternative last exons (AL) and mutually exclusive exons (MX) groups.

**Figure 4 genes-11-01300-f004:**
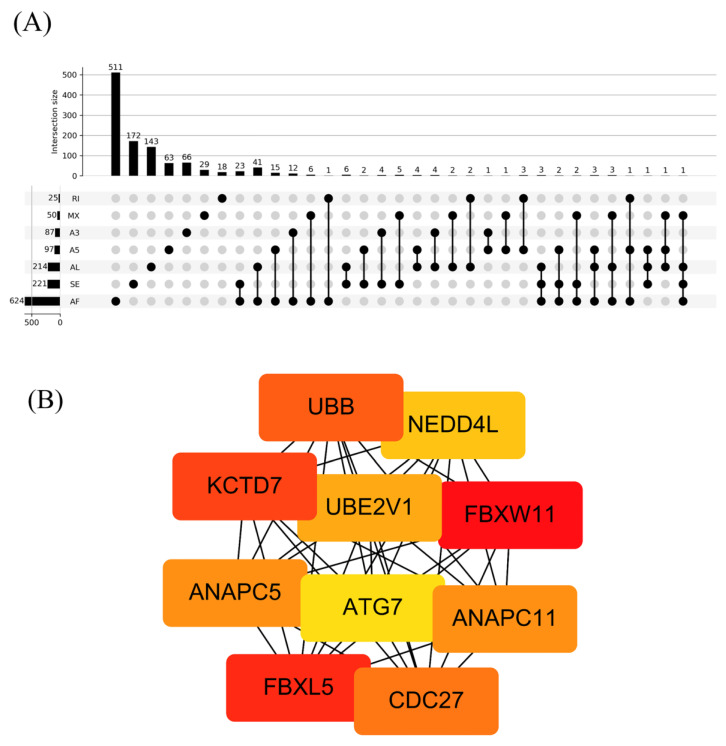
Summary and gene network of OS-related AS events. (**A**) UpSet intersection diagram showing the seven types of OS-related AS events. (**B**) Hub gene screening.

**Figure 5 genes-11-01300-f005:**
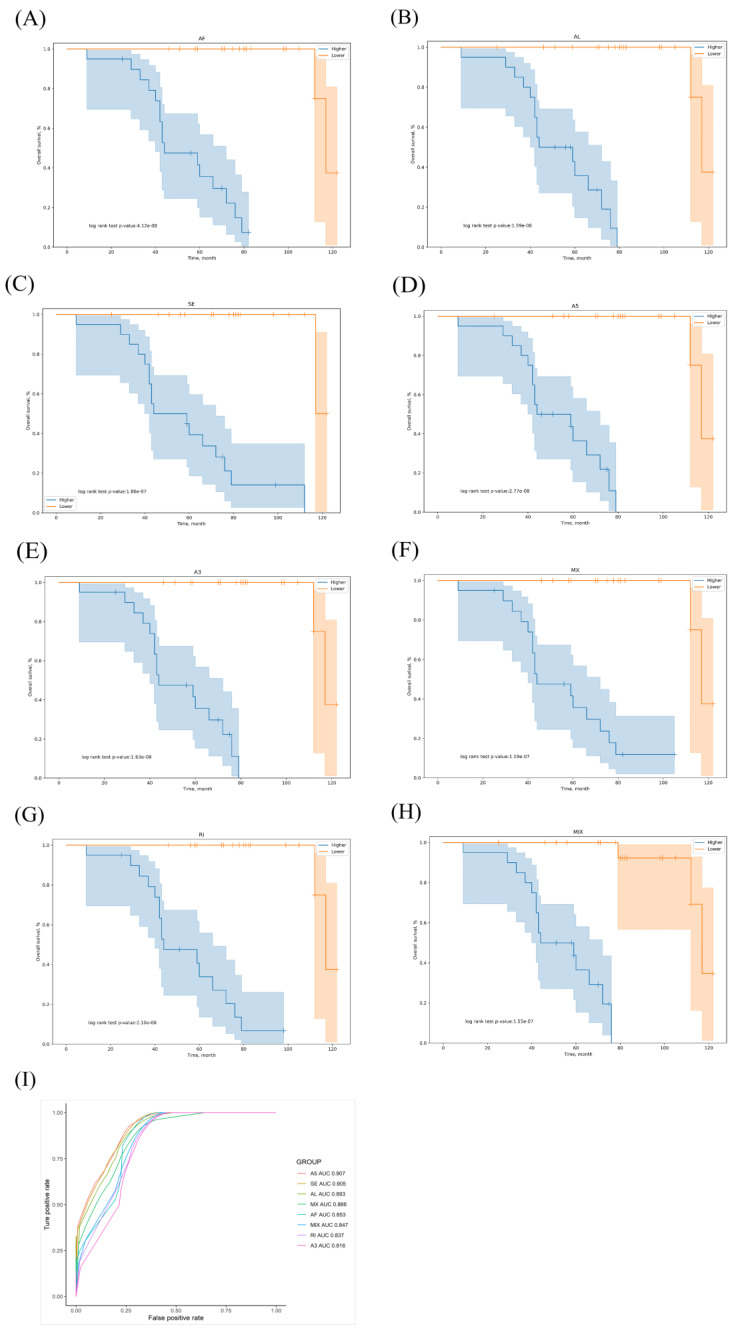
Kaplan–Meier and receiver operating characteristic (ROC) curves of the prognostic predictors in our LUAD cohort. (**A**–**G**) Kaplan–Meier curves of the prognostic predictors constructed for each type of AS event. Blue line indicates high-risk group; red line indicates low-risk group. (**H**) Kaplan–Meier curves of the mixture (MIX) prognostic predictor constructed for all types of AS events. (**I**) ROC analysis for all prognostic predictors. The colored lines of the ROC curves of the prognostic predictors represent different types of AS events.

**Figure 6 genes-11-01300-f006:**
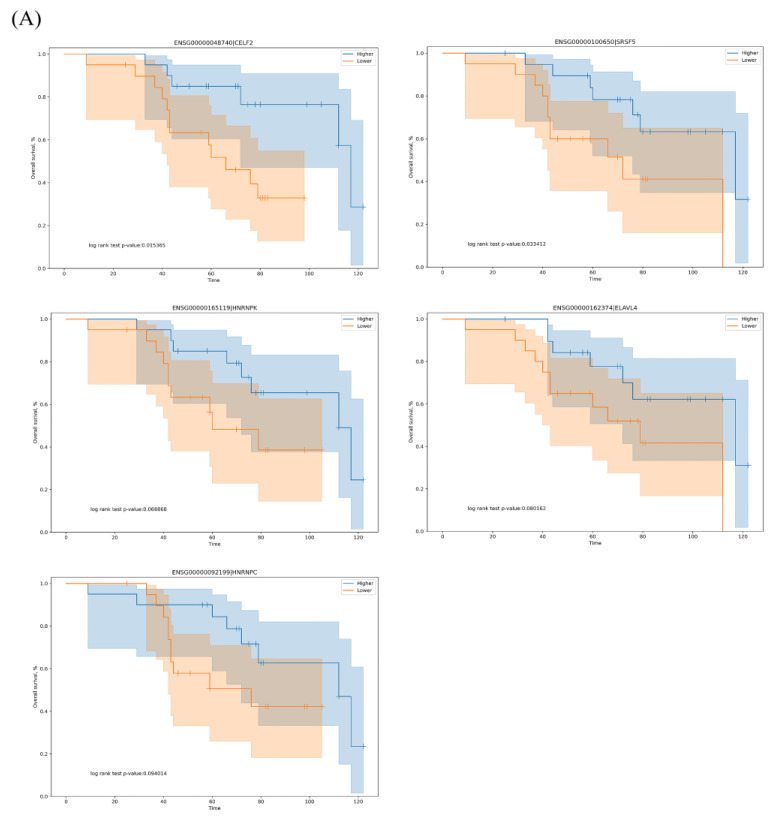
The correlation network of splicing factors (SFs) and the top 20 OS-related RI, SE, A5, A3, AF, AL and MX events in our LUAD cohort. (**A**) Kaplan–Meier curves of the *CELF2*, *SRSF5*, *HNRNPK*, *ELAVL4* and *HNRNPC* SFs. Blue line indicates the high-expression group; red line indicates the low-expression group (based on median values). (**B**) AS correlation network constructed using Cytoscape. Five OS-related SFs were positively (red lines) or negatively (blue lines) associated with AS events. Red and blue dots indicate favorable and adverse AS events, respectively. A circle represents one type of AS. (**C**) Correlation analysis between the expression of *HNRNPK* and the RI PSI values of *MED16*, between the expression of *SRSF5* and the AF PSI values of *ZNF385B* and between the expression of *HNRNPK* and the AL PSI values of *ASPH*. blue line: regression line; blue shaded part; confidence interval for the regression estimate; blue circle: sample

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
