# Peer review of "Genome-Wide Analysis of Prognostic Alternative Splicing Signature and Splicing Factors in Lung Adenocarcinoma"

_genes, 2020, doi:10.3390/genes11111300_

Round 1

Reviewer 1 Report

The study by Chang et al. reports on the identification of prognostic markers in lung adenocarcinoma (LUAD) using RNA-seq data obtained from the analysis of tumor tissues in 40 patients. Alternative splicing (AS) events that have clinical significance were specifically searched for. Different bioinformatics and statistical analyses were performed to reveal correlations between overall survival and AS events and/or the expression level of splicing factors. Overall the study mostly identified favorable prognostic factors. This is mainly a descriptive study that provides interesting preliminary data on prognostic factors for LUAD that will need to be validated on larger cohorts. It would have been useful to have a number of experimental validations to reinforce the results.

A number of specific comments need to be addressed by the authors:

  • Concerning the SUPPA2 analyses: Alternative first exon (AF) event is by far the most frequent type of splicing event detected here (48% of the detected AS events), which does not reflect the frequency of AS categories generally detected (exon skipping is the most frequent AS events). The data from this study also differ from those reported in the TGCA cohort in Figure 2B where SE are the most common AS events reported. Could there be any bias in the analysis/reference transcripts used?
  • A5 events were found to be more effective for distinguishing the survival outcome of LUAD patients than the predictor models built using the other six types of AS pattern. Could a gene ontology analysis or cytoscape analysis be useful for those genes which exhibit such alternative splicing (use of a 5’ alternative splice site) to try to find a functional correlation?
  • The identified splicing factors whose expression level is modified (ELAVL4, CELF2, HNRNPK, HNRNPC, SRSF5) differ from those previously reported to be deregulated in lung cancer (ex: SRSF1) in the cited reference (Coomer et al. 2019). The authors could discuss further their results in relation to the published results. This study showed that high-expression levels of the 5 identified splicing factors are associated with better prognosis. However, the sentence “Our study provides a further understanding of splicing patterns and their mechanisms associated with SFs in LUAD, which may elucidate the underlying mechanisms of AS in the oncogenesis of LUAD” seems to be over-interpreted as there is not experimental data to demonstrate that CELF2 or SRSF5 identified splicing factors play a role in the detected alternative splicing events. The functional link is not established.
  • The authors acknowledged the limitations of their study. Even if experiments have not been performed to better characterize the functional consequences of the identified AS events, it would have been important to validate the more significant AS events detected by RNA-seq data analyses by independent RT-PCR experiments (for example splicing isoforms of ABHD14A).
  • The writing in the circles in figure 6B is not readable.
  • The authors indicate that “the CDKN2A_ENST00000497750 isoform may have a stronger tumor suppressor function than the wild type transcript”. A more detailed description of the AS (A5) in the CDKN2A gene, impact on the reading frame, change in coding sequence in relation to functional domains would be useful to support the assumption.
  • The conclusion “we suggest that the hub genes (FBXW11, FBXL5, KCTD7, UBB, and CDC27) may be potential targets for the prevention and treatment of LUAD in the future” could be tone-down.

Reviewer 2 Report

In their manuscript, Chang et al. look for splicing and gene expression changes that are associated with overall survival in lung adenocarcinoma. They identify a number of variations of all splicing classes associated with differential overall survival. Likewise, they identify splicing factors whose expression is associated with overall survival. Although I do not dispute the validity of the findings, I do have a number of concerns about the manuscript as it is.

Main comments:

  • It is not clear how the authors define what they consider to be an alternative splicing event. AS is always relative to something else. Are they looking as splicing events that vary in PSI between LUAD samples (in which case the events would be better referred to as “splicing variations”), or are they looking at events that are differentially spliced between LUAD and healthy tissue (in which case, what is the reference healthy tissue?) or are they looking at unannotated, previously unreported splicing variations that they are able to observe in LUAD?

  • How many splicing events are initially selected for the analysis in the section 3.1? 1957 events are associated with OS, but how many total events were included in the analysis and how were they selected (again, related to the 1st comment)?

  • Multiple testing correction should be applied on the Cox regression analysis presented in section 3.1. It is not clear whether this was applied or not on the results that are presented. 1957 significantly associated events seems like rather high number.

  • In section 3.1, the authors write that “most of the OS-related AS genes underwent at least two types of AS” but figure 4A clearly shows that a large majority of genes harbor a single splicing event.

  • In figure 3 each panel should be labeled with the type of splicing event represented.

  • In section 3.3 it is not clear how splicing factors were selected for analysis. How many SFs were included in the analysis and how were they selected? Again, was multiple testing correction applied?

  • In section 3.3, are any of the top OS-associated SFs frequently mutated in LUAD?

  • The font size for many, many labels and axes need to be dramatically increased. Figure 3 and 6B would be better served by accompanying tables.

  • The correlations shown in Figure 6C are remarkably poor. The clear-cut binary distribution of PSI values for the 3 events shown in 6C is also most unusual but bulk RNA-Seq of any tissues and makes me wonder about the validity of the PSI quantification method. The y-axis should also be limited to a [0,1] interval given that we are looking at PSI values.

Minor points:

  • The ending sentence of the abstract is weak/non-specific.
  • The manuscript would benefit from a thorough check for typos and syntax.

Round 2

Reviewer 1 Report

I thank the authors for their prompt reply and the changes they have made in the manuscript. Page 7, the authors may consider to make the minor change below, as ADA2 is not indicated in Figure S1. "We found that NRP2 and ADA2 (CECR1) were involved in the heparin-binding pathway (Figure S1)."  

Reviewer 2 Report

The fact that no correction is applied for multiple testing means that any/all of the correlations and associations observed throughout the study could simply be noise. The fact that only 4 splicing variations are still significant after multiple testing correction, as stated in the response to comment #3, strongly points in that direction. Consequently, at the very least, a statement that no correction was applied for multiple testing in the Cox regression analyses should be added to section 2.6 of the methods before publication. This would give a chance to the readers to decide where they stand relative to the interpretation of the results.
